# Whole-Genome DNA Methylation Analysis in Hydrogen Peroxide Overproducing Transgenic Tobacco Resistant to Biotic and Abiotic Stresses

**DOI:** 10.3390/plants10010178

**Published:** 2021-01-19

**Authors:** Ana L. Villagómez-Aranda, Luis F. García-Ortega, Irineo Torres-Pacheco, Ramón G. Guevara-González

**Affiliations:** 1Biosystems Engineering Group, Engineering Faculty, Amazcala Campus, The Autonomous University of Querétaro, Highway Chichimequillas s/n Km 1, Amazcala, El Marqués, Querétaro C.P. 76265, Mexico; annvillaranda@gmail.com (A.L.V.-A.); torresirineo@gmail.com (I.T.-P.); 2Department of Genetic Engineering, Center for Research and Advanced Studies of the National Polytechnic Institute (Cinvestav), Highway Irapuato-Leon, Km 9.6, Libramiento norte, Irapuato, Guanajuato C.P. 36821, Mexico; luis.garcia@cinvestav.mx

**Keywords:** DNA methylation, hydrogen peroxide, hypomethylation

## Abstract

Epigenetic regulation is a key component of stress responses, acclimatization and adaptation processes in plants. DNA methylation is a stable mark plausible for the inheritance of epigenetic traits, such that it is a potential scheme for plant breeding. However, the effect of modulators of stress responses, as hydrogen peroxide (H_2_O_2_), in the methylome status has not been elucidated. A transgenic tobacco model to the *CchGLP* gene displayed high H_2_O_2_ endogen levels correlated with biotic and abiotic stresses resistance. The present study aimed to determine the DNA methylation status changes in the transgenic model to obtain more information about the molecular mechanism involved in resistance phenotypes. The Whole-genome bisulfite sequencing analysis revealed a minimal impact of overall levels and distribution of methylation. A total of 9432 differential methylated sites were identified in distinct genome regions, most of them in CHG context, with a trend to hypomethylation. Of these, 1117 sites corresponded to genes, from which 83 were also differentially expressed in the plants. Several genes were associated with respiration, energy, and calcium signaling. The data obtained highlighted the relevance of the H_2_O_2_ in the homeostasis of the system in stress conditions, affecting at methylation level and suggesting an association of the H_2_O_2_ in the physiological adaptation to stress functional linkages may be regulated in part by DNA methylation.

## 1. Introduction

Plants are regularly exposed to stress factors, and due to their sessile nature, they have developed sophisticated defense responses at a physiological, biochemical, and genetic level. It is well known that the environment and the genotype give the phenotype of an organism. This is partly due to the epigenetic phenomena, where modifications in DNA, RNA, and histones can alter the genetic expression patterns of genes. The environment determines these modifications; they are labile but can be stable and heritable [1,2]. Therefore, the epigenetic modifications are part of the genetic plasticity of organisms [3] and in plants are also essential in acclimatization (stable during the same generation) and adaptation processes (conservable to subsequent generations) [4].

The DNA modifications include C5-methylcytosine (5mC), N6-methyladenine (6mA), and N4-methylcytosine (4mC) [5]. There are some other DNA modifications as the O6-methylguanine (O6-mG), N7-methylguanine (7mG) or N3-methyladenine (N3mA) that are believed to be cause by endogenous and/or environmental alkylating agents [6] and are emerging as important markers with adverse effects. 5mC methylation is the most abundant and studied. As mentioned, in plants, the 5mC presence ranges from 5 to 30% depending on the species [7], while others, as the 6mA levels are less than 0.8% in eukaryotes and in *Arabidopsis* has been reported as less than 0.2% [5]. In plants, the 5mC DNA methylation exists in three different sequence contexts: CG, CHG, and CHH. The CG context takes place in dinucleotides C and G, and it is called symmetrical context. For the CHG and CHH contexts, the H can be any base as A, T, or C [8,9]. The accumulation of DNA methylation in all contexts results in chromatin condensation and, therefore, gene silencing [8].

DNA methylation is involved in the replication and mismatch repair, the control in gene expression, tissue-specific gene expression, transcript synthesis, signaling to host-pathogen interaction, virus resistance, gene imprinting, and epigenetic memory [5]. DNA methylation is a highly stable epigenetic mark, due to it can be maintained during the mitotic and meiotic cell division processes, such that these changes provide a mark plausible for the inheritance of metastable epigenetic traits [9,10,11]. In plants, DNA methylation alterations occurs as part of the response to stress factors as drought [12], salinity [7,13], cold-stress [14], heat-stress, presence of heavy metals [15], nutriment deficit [16,17], wounding [18], viral infection [19] pathogen attack [20] and even spaceflight [21]. Similarly, the priming with some compounds alters the DNA methylation behavior of the plant. β-Aminobutyric acid (BABA) [22] and methyl jasmonate (MeJA) [23] induce DNA hypermethylation, and it is suggested that it may be associated with the development of memory to enhance stress tolerance. Due to this nature of the epigenetic changes, they are a potential strategy for plant breeding to obtain more adaptable crops to stress. Further, it is necessary to elucidate what kind and doses of stress factors can induce an epigenomic response in the plant genome.

Hydrogen peroxide (H_2_O_2_) is an essential compound in stress perception and signaling. It interacts with the Ca^2+^/K^+^ channels, phytohormones such as abscisic acid (ABA) and Salicylic acid (SA), MAP kinases, miRNAs, and transcription factors that lead to the activation of stress-related genes [24,25,26]. It is considered a modulator of the stress response and immunity pathways in plants since the exogenous application of H_2_O_2_ can improve the responses to stress [27] as shown in several studies with a positive effect on tolerance to saline stress [28,29,30,31], drought [32,33], high temperatures [34,35], flooding [36], virus infection [37,38] and low lighting conditions [39].

Our research group has a model to study the effects of H_2_O_2_ in plant stress responses: a transgenic *Nicotiana tabacum* cv. Xanthi line overproducing H_2_O_2_ constitutively. This is due to the expression of the transgene *CchGLP* of *Capsicum chinense* Jacq, which encodes for an Mn-superoxide dismutase (Mn-SOD). The *CchGLP* gene is involved in the natural resistance to geminivirus in accession BG-3821 of *C. chinense* from Yucatán, México [40]. The expression of CchGLP in *Nicotiana tabacum* xanthi nc resulted in several transgenic lines, of which two lines stood out: Line 8 (L8) with a high gene expression and high H_2_O_2_ production; and Line 1 (L1) with low gene expression and low H_2_O_2_ production, which is considered as azygous control. Both transgenic lines presented a characteristic phenotype against biotic and abiotic stress of resistance and susceptibility, respectively, to geminivirus infection [41], drought [42], and heavy metals [43]. In previous works, different omic profiles were determined to elucidate the molecular elements involved in the resistance phenotype of the transgenic line at different molecular levels: proteomic [44], metabolomic [44], transcriptomic [42], and miRNAs profiles [43]. The L8 has shown a higher occurrence of metabolism and cell growth and differentiation-related proteins, and metabolites related to phenylpropanoid metabolism, osmotic process, and amino acid metabolisms [42]. Likewise, the differential transcriptomic between these two lines displayed genes associated with primary metabolism and cellular functions as cell wall reinforcement and secondary metabolism as phenylpropanoids, terpenoids, alkaloids, and steroid hormone biosynthesis [44,45]. In the case of the miRNAs differential profile, the results were associated mainly with the structural molecular activity and proton-transporting type ATPase [43].

It has been suggested that epigenetic marks may have a connection with signaling reactive oxygen species (ROS) and redox metabolism, due to it has been observed that the enzymes responsible for histone methylation are sensitive to the production of ROS [46,47]. However, no studies have been conducted on the effect of H_2_O_2_ on epigenetic mechanisms in plants. Therefore, the present study aimed to determine changes in the 5-methylcytosine DNA methylation status of the transgenic tobacco H_2_O_2_ overproducing line (L8), which has resistance to biotic and abiotic stresses in comparison with the azygous stress-susceptible control (L1). This knowledge could be a fundamental insight into the role of the epigenetic plant strategies used in the adaptation to stress conditions, and the role of the H_2_O_2_ in the physiological adaptation, with the expectation of being applied to enhance the plant tolerance to changing environments.

## 2. Results

### 2.1. Phenotypic Characteristics

L8 (overproducing H_2_O_2_) and L1 (azygotic line) have typical growth under the experimental conditions used, as shown in Figure 1. L8 displayed a slight growth delay compared with L1, likely associated with the energy demands of its phenotype. However, there were no significant differences in height, fresh weight, foliar area, stem diameter, and chlorophyll content, as described in Table 1. Therefore, as mentioned in previous works, the H_2_O_2_ overproduction levels in L8 do not significantly affect the growth and morphology of plants during their life cycle.

### 2.2. DNA Methylation Levels and Distribution

To characterize transgenic tobacco methylomes, we generated single-base resolution maps of DNA methylation by whole-genome bisulfite sequencing (WGBS) for 6-week-old plants of both transgenic tobacco lines. Each transgenic line was sequenced with three biological replicates, and every sequencing library was produced and sequenced. It was obtained an average of 461205823 clean reads by each sample, with quality score of Q20 average of 97.73% (Appendix A). The methylomes presented a mapping efficiency with the reference genome in a range between 55 and 71% of unique alignments per library (Appendix A). The principal methylation context was CG, in both transgenic plants, with 84.6% for L8 and 85.4% for L1 (Figure 2A). Also, the lesser abundant context was CHH with 15.7% and 14.1%, respectively. This represented an average methylation level of 48.64% for CG, 42.33% on CHG, and 9.03% on CHH of the total DNA methylation levels in the transgenic line L8 (Figure 2B). However, the general methylation levels and context distribution were not changed noticeably between transgenic lines in normal growth conditions.

### 2.3. Distribution of Differentially Methylated Cytosines

To examine differences in DNA methylation between the two transgenic line, we compared the cytosines methylated between overproducing (L8) and azygous (L1) plants to define the differentially methylated cytosines (DmCs) as hyper or hypomethylated, according to the methylation proportion of a cytosine between the L1 and L8 samples at the same site. The hypermethylated and hypomethylated cytosines were distributed similarly across the chromosomes of the genome (Figure 3A). The major abundance of hypermethylated cytosines was mapped in chromosomes 11 and 16, while the hypomethylated was in chromosomes 9 and 12.

In total, 9432 DmCs were found, among which 4765 were hyper-methylated, and 4667 were hypomethylated. The higher proportion of DmCs was identified in the CG context, and the lower sites were observed in the CHH context (Figure 3B). The latter was consistent with the global distribution of DNA methylation patterns of tobacco plants. However, the DmCs distribution in different genomic elements (genes, promoters, introns, intergenic regions, and transposable elements) was variable in each context (Figure 4A–F). The intergenic regions were the most enriched element with methylation in almost all contexts, except by the hypermethylated CHG and CHH contexts. In the CG context, the intergenic regions methylation was 91.34% and 92.5% for hyper and hypomethylated DmCs, respectively. In the case of the hypomethylated DmCs in CHG and CHH on intergenic regions were 69.8% and 79.3%. The intron regions were significantly abundant in the hypermethylated DmCs in CHG and CHH contexts, with 43.63% and 52%, respectively.

In order to identify direct linkages between the DmCs found and the phenotype of the overproducing H_2_O_2_ transgenic tobacco plants, we focused on the DmCs within the protein-coding regions. The higher DmCs on exons was observed in the CHG context (Figure 4B,E), in which 360 DmCs hypermethylated and 636 DmCs hypomethylated were found. By contrast, in the CG context, 107 DmCs hypermethylated and 105 hypomethylated were detected. For the CHH context, just three DmCs hypermethylated and 23 DmCs hypomethylated were identified. The data of DmCs identified on exons are listed in Appendix A.

### 2.4. Gene Ontology Analysis

The gene-ontology analysis applied to the DmCs in protein-coding regions showed enrichment in hypomethylated DmCs on respiratory and energy metabolism (Appendix A). In the category biological process, the functions identified were photosynthetic electron transport chain, photosynthesis, photosynthetic light reactions, generation of precursor metabolites and energy, electron transport chain. In the category molecular function, the hypermethylated DmCs were associated with enrichment for the ADP binding process.

### 2.5. Correlation of Differentially Methylated Cytosines and Differential Gene Expression

The correlation of genes with DmCs and differential expressed Genes (DEGs) was analyzed to determine the biological relevance of the methylome in the physiological adaptation to stress. For this analysis, 1117 DmCs sites were considered, including the sites on protein-coding regions and promoter regions. Also, there were considered the data set of DEGs [42] and small-RNA targets (sRNA) [43] obtained in previous works, considering that the same grown conditions were used in the experiments and the physiological age of plants tested were identical. Of the genes differentially expressed in response to H_2_O_2_ in the transgenic tobaccos, 5397 DEGs were considered, and among those, 83 genes were also differentially methylated (Figure 5A). Most of the genes were methylated in the gene body in the CHG context, where they tended to hypomethylation, as mentioned in previous sections (Figure 5B).

Figure 5 shows a heatmap of 83 genes with DmC-DEG, where the gene ID number and identity of the gene was aligned with the differential methylation level and the context in each one (Figure 5C), as well as the differential expression (Figure 5D). In most of the genes, a positive correlation was observed, where hypomethylated sites were associated with downregulation expression of genes.

Interestingly, three sRNA genes in the L8 transgenic tobaccos were identified, corresponding to an abortive infection protein, a cytidylyltransferase, and a terpene synthase; and one in the L1 tobaccos matched with a gelsolin. 

## 3. Discussion

The present study provides insights into the study of H_2_O_2_ roles with the DNA methylation and their impact on the physiological adaptation to stress factors. The data has high reliability and accuracy. The mapping efficiency obtained is comparable to the informed in other WGBS studies [16,48,49,50]. Likewise, the proportion of the three-methylation context in the transgenic tobaccos was similar to those reported in other plant models as soybean [51], barrel clover [7], maize [16], *Arabidopsis* [21,50], tomato [52], burberry [48] and watermelon [49], among others. In all these studies, the higher methylation levels were located in the CG context. Similar results in the distribution of DmC in different genomic regions were obtained by Manoharlal et al. In their study presents a methylome-transcriptomic profile of tobacco after gibberellin A3 application [53]. The high number of methylations in intergenic regions may be in part due to the high content of repetitive sequences in the tobacco genome. Thus, part of the differences in the proportion of hypermethylated and hypomethylated regions in their study and ours may be due to the variable of treatment and the transgene, respectively.

Nevertheless, in our data, the most changes in the DmCs sites were in the CHG context. This pattern was also reported in other DNA methylation studies related to stress responses, as salt stress [54]. There has been pointed out that the CHH hypomethylation had a high relevance in plant defense, and it is related to the RNA-directed DNA methylation pathway [55]. Despite the low number of sRNAs correlated with the WGBS data, some genes DmC-DEG were associated with RNA binging and RNA metabolism. Also, it is possible that exists other sequences of non-coding RNAs not identified among the sites with differential methylation on intergenic regions, which was the most abundant in this study. The non-coding RNAs play a critical role in the transcriptional and posttranscriptional regulation of gene expression and lately have emerged as key regulatory elements in the plant stress responses [56].

It is pointed out that the H_2_O_2_ can induce stress tolerance in plants when applied exogenously, and due to several redox-sensitive genes are involved in stress acclimation [27]. Similarly, recent evidence indicates a close relationship between the ROS regulation and the epigenetic regulation in a negative association. In several studies has been suggested that ROS accumulation causes DNA hypomethylation [47,57,58]. Our data agree with these assumptions, considering that most DmCs were hypomethylated. It suggested that the H_2_O_2_ overproduction provokes a new methylation status in the genome, according to the necessities of the system to reach a balance. Due to this, the functional linkages pointed out by the GO analysis categories were photosynthesis and energy metabolism as the most affected by the DmCs on genes in response to the overproduction of H_2_O_2_. Some genes identified that also exhibited an up-regulation in DEG were associated with proton transport, as the V-type proton ATPase and the ABC transporter-like. These data corroborate that H_2_O_2,_ as part of the ROS balance, plays a crucial role in the physiological adaptation of plants, such that the plant under continuous oxidative stress has reached a balanced state where the ROS metabolism has been adjusted through the respiration and energy processes.

Additionally, it has been demonstrated that H_2_O_2_ has a strong link with calcium signaling on gene expression regulation, antioxidant defense, and development modulation. Crosstalk among them is given in stress responses when H_2_O_2_ can activate calcium channels and calcium-dependent protein kinases. [59,60,61]. Furthermore, H_2_O_2_ can play a key function on DNA methylation for the activation of genes related to calcium as it is suggested by the presence of DmC-DEG in genes as the calcium-binding protein 39, the gelsolin, calmodulin-binding heat-shock protein, calcium-dependent protein kinase. Calcium participates in the cell signaling pathways, and it is considered as a sensor to external stimulus. Moreover, some DmC-DEG genes were associated with membrane transport, as the adaptor protein complex involved in the protein transport and related to the recruitment of clathrins and recognition of cargo molecules. In the study of Atighi et al. it was suggested that DNA hypomethylation is part of the general plant pattern-triggered immunity defense responses due to their inverse association with susceptibility to nematodes [62], such that there was a strong relationship with the recognition, molecule transport, and signaling process and changes in DNA methylation.

In the study performed by Cao et al. it was reported that H_2_O_2_ pretreatment-triggered DNA methylation variations, which may alleviate methylation and damage induced by heat stress in cucumber. Even when this analysis was limited to MSAP loci, it emphasized the relevance of H_2_O_2_ to enhance or inhibit gene transcript levels by the effect methylation changes [34]. Among the genes DmC-DEG associated with stress responses, the legume lectin, the armadillo-like protein, and SNARE11 were identified in our data. Also, two genes related to the methylation process (5-methyltetrahydropteroyltriglutamate -homocysteine methyltransferase and histone-lysine N-methyltransferase) and three associated with DNA repair (ATMRK serine-threonine protein kinase-like, Telomerase activating protein, DNA mismatch repair protein) were found. Also, the presence of some genes associated with RNA binding proteins and nucleic acid binding proteins suggested that the H_2_O_2_, despite related to DNA damage, the continuous H_2_O_2_ levels in the plant activate the mechanisms to protect the genetic material of the individual.

Although large-scale changes in the DNA methylation status, and either a wide number of genes stress-related within the genes with DmC and DEG were not found, some changes in the protein-coding regions with DmCs identified in the study. In our case, there was a low coincidence with the transcriptomic data due to the different depth coverage of the omics data obtained in each profile. Further studies will be carried out to confirm the expression of some of these genes in the transgenic plant, to validate the data obtained by WGBS.

Taken together, the data obtained in the present study strongly suggest that the H_2_O_2_ overproduction in the transgenic tobacco could be associated with rapid recognition of stress and triggering of defense responses, which may be critical in the well-timed activation of the defense system and by consequence, in the physiological adaptation to stress. This is related to the study of Zhou et al., in where spaceflight induced a significant number of genes DmC-DEG associated with ROS signaling, demonstrating that ROS genes play a fundamental role in the physiological adaptation of plants to stress [21]. Considering that the spaceflight environment is a unique and unconventional stress condition, they highlighted the importance of the ROS signaling in the shaped responses to stress, which made them a critical point to stimulate the adaptation of the crops to a more changing and unpredictable environment. Finally, the whole-genome differential cytosine methylation found between the two transgenic tobacco lines helps explain the differential phenotype related to stress response displayed by these plants; moreover, suggesting that an epigenetic component at cytosine DNA methylation it is important in the phenomenon.

## 4. Materials and Methods

### 4.1. Transgenic Plants Growth Conditions

Seeds from the transgenic lines L8 (overproducing H_2_O_2_) and L1 (acigotic line) obtained in previous works were used [41]. The seeds were surface sterilized by immersion in 10% sodium hypochlorite solution for 5 min and washed three times with sterile distilled water. The seeds were left in sterile water for three days; then 20 seeds of each line were sowed in flasks with Murashige-Skoog medium with 100 µg/mL of kanamycin as a selection agent for the transgene. They were kept at room temperature under a 16 h light/8 h dark photoperiod. After three weeks the seedlings were transferred into individual pots with Mixture 3 Peat Moss Sunshine as substrate and kept inside a glass chamber at room temperature with humidity of 70% for three days. They were kept in a greenhouse until seed obtaining. The plants were watered daily.

### 4.2. Tissue Sampling and Physiological Testing

Three 6-weeks-old plants of each group were selected randomly for sampling. The leaves of each plant were cute and frozen in liquid nitrogen and stored at −70 °C. For the physiological analysis, 6-weeks and three-month-old plants were measured for height, fresh weight, dry weight, stem width, foliar area, and chlorophyll content approximation by a SPAD meter.

### 4.3. DNA Extraction

Genomic DNA was extracted from 200 mg of leaf tissue of each sample, which was pulverized with liquid nitrogen. Extractions were carried out by the cetyltrimethylammonium bromide (CTAB) method [37]. DNA purity and concentration were measured in a Nanodrop spectrophotometer (Accesolab, Mexico City, Mexico), and DNA quality was verified by electrophoresis using 1.2% agarose gel stained with RedGel.

### 4.4. WGBS

A total of six DNA samples (three individuals per group) were sent for whole-genome bisulfite sequencing (WGBS) to the Beijing Genomic Institute (BGI, Beijing, China). The library preparation was done according to the BGI standard directional WGBS pipeline. Genome fragmentation into 100–300 bp fragments was done by sonication. Bisulfite treatment was done with the EZ DNA methylation Gold Kit (ZYMO, Mexico City, Mexico). The WGBS consisted of a 150 bp paired-end sequencing with a genome coverage of 12× on a NovaSeq 6000 system (Illumina, Inc., San Diego, CA, USA).

### 4.5. Data Processing

Raw reads were pre-processed with the Fastp v0.20.0 [63] using −c −x −y options for polyX trimming in 3′ ends, base correction in overlapped regions, and low complexity filtering, respectively. Clean reads were aligned to the *N. tabacum* reference genome (EBI: AWOJ01000000) using Bismark aligner v0.22.3 [64] under the parameters −N 1 −L 20 –bowtie2 -nondirectional. The resulting BAM files were sorted and indexed using Samtools v1.9 [65]. Subsequently, the differential methylation analysis was performed using Methylkit v1.14.2 [66] in R v4.0.0 [67] as follows: Methylated cytosines were extracted from aligned reads using processBismarkAln function. To filter bases above the background expected from inefficiencies in the bisulfite conversion reaction and sequencing errors, the filterByCoverage function was used to discard bases that have coverage below 10X and discard the bases that have more than 99.9th percentile of coverage in each sample. The parameters for the calculated DiffMeth function were slim = TRUE, weighted. mean = TRUE. Differential methylated cytosines (DmC) were defined as cytosines that had q-value < 0.01 and methylation difference > 25% between L1 (control) and L8 (test) methylation rates. Only was considered DmC present in the 3 biological replicates to made more rigorous our methylation changes detection. Gene ontology enrichment analysis was performed with R package clusterProfiler [68], selecting significant GO categories with q-value < 0.05.

### 4.6. Correlation between DmC and DEGs

For analyzing the relationship between the differential methylated cytosines (DmC) obtained in this study and the genes with differential expression (DEGs) obtained in previous work [42]. The data was graphic with R package heatmap.

### 4.7. Statistics and Data Visualization

The graphs were generated with R software. An ANOVA analysis was performed in R using the ANOVA function in the stats package and Duncan’s multiple range test.

### 4.8. Data Availability

Bisulfite sequencing data reported in this study are available under the GSE161166 series accession number at the Gene Expression Omnibus (GEO) database.

## 5. Conclusions

H_2_O_2_ overproduction in the transgenic tobacco plants has a high impact on the cytosine DNA methylation status of associated genes, especially with CHG context changes on the gene body regions. The impact was observed in energy metabolism genes, molecule transport, sensing, and signaling related to calcium and DNA repair. This work corroborates that the H_2_O_2_ is a key compound in the biological process, with a molecular significance at an epigenetic level.

## Figures and Tables

**Figure 1 plants-10-00178-f001:**
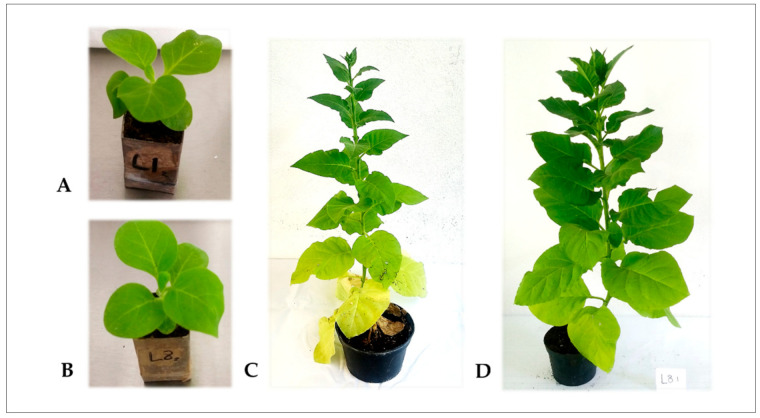
Phenotype of transgenic tobacco plants. In the photos, there is shown the phenotype characteristics of the transgenic plants in two physiological ages. The first, at 6-weeks-old both plants (**A**) L1 and (**B**) L8 were practically identical. At this age, samples were taken for the WGBS. At 3-months-old, both plants still looked similar, but the plants (**C**) L1 presented a light yellowing in the basal leaves compared with the (**D**) L8 plant.

**Figure 2 plants-10-00178-f002:**
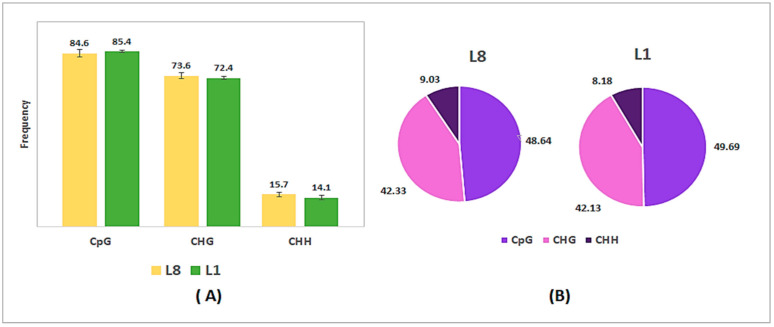
DNA methylation levels in high (L8) and low (L1) H_2_O_2_ producing transgenic tobacco plants by WGBS. (**A**) Global methylation pattern of transgenic tobacco plants and their (**B**) Proportion of each context in the total methylation in both lines.

**Figure 3 plants-10-00178-f003:**
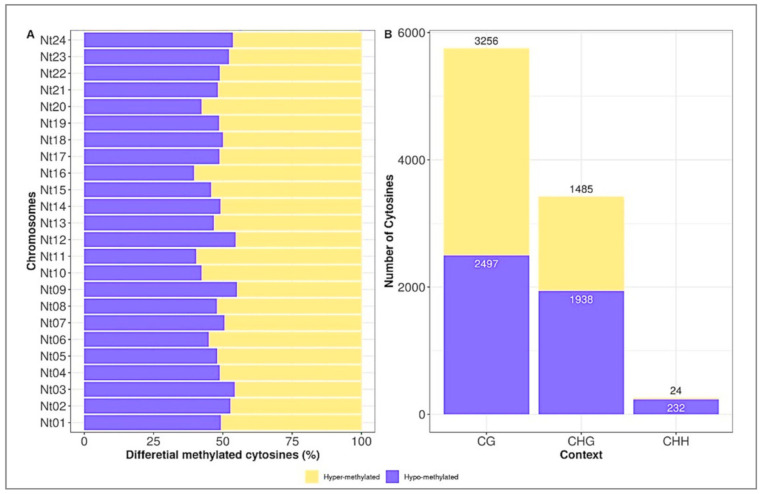
Distribution of relative DmC levels for each chromosome (**A**) and CG, CHG and CHH context (**B**) in the high H_2_O_2_ producer transgenic tobaccos.

**Figure 4 plants-10-00178-f004:**
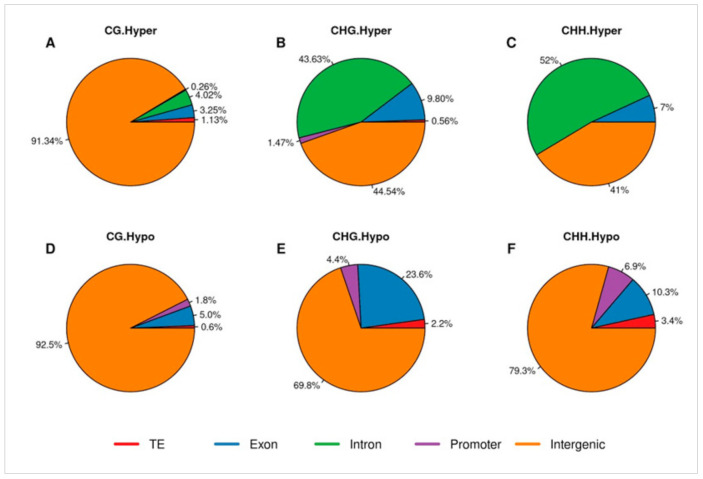
Distribution of genomic elements with hypo-DmCs and hyper-DmCs in the whole genome for each context. (**A**) hyper-CG. (**B**) hyper-CHG. (**C**) hyper-CHH. (**D**) hypo-CG. (**E**) hypo-CHG. (**F**) hypo-CHH. Transposable elements (TE) in red colour; exons in blue colour; introns in green colour; promoter region (2 kb upstream of TSS and 2 kb downstream of TES) in purple colour; intergenic regions in orange colour.

**Figure 5 plants-10-00178-f005:**
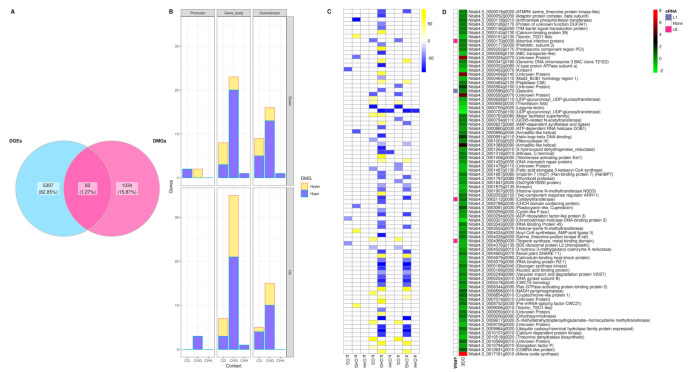
Correlation of differential methylation and differential gene expression in transgenic tobacco associated with the H_2_O_2_ response. (**A**) Number of DMCs associated with DEGs. (**B**) DMCs profiles of the up/down-regulated genes in different sequence contexts and gene regions. Scale color represents methylation level difference (blue = hypomethylation; yellow = hypermethylation). (**C**) DmCs profiles of up/down-regulated genes involved in H_2_O_2_ response in tobacco. (**D**) Gene expression profiles (L8/L1) of the genes with DmCs. Scale color represents log2 fold-change expression (red = upregulation, green = downregulation).

**Table 1 plants-10-00178-t001:** Physiological features.

Feature	L8 (6 Weeks)	L8 (3 Months)	L1 (6 Weeks)	L1 (3 Months)
Height (cm)	6.4 ± 0.51	72 ± 3.46	7.1 ± 0.24	74 ± 1.0
Fresh weight leaves (g)	0.55 ± 0.03	41.6 ± 4.19	0.57 ± 0.02	56.9 ± 3.43
Fresh weight root (g)	0.12 ± 0.02	25.9± 5.23	0.13 ± 0.01	28.2 ± 2.68
Stem width (mm)	1.54 ± 0.11	11.3 ± 0.12	1.52 ± 0.14	10.8 ± 0.22
Leaf area (cm^2^)	47.5 ± 3.7	1775 ± 313	49.1 ± 2.9	2607 ± 134
SPAD value	39.4 ± 2.09	46.7 ± 2.24	38.2 ± 2.14	44.8 ± 2.55

## Data Availability

Bisulfite sequencing data reported in this study are available under the GSE161166 series accession number at the Gene Expression Omnibus (GEO) database.

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
