# Peer review of "Whole-Genome DNA Methylation Analysis in Hydrogen Peroxide Overproducing Transgenic Tobacco Resistant to Biotic and Abiotic Stresses"

_plants, 2021, doi:10.3390/plants10010178_

Round 1

Reviewer 1 Report

In this paper, Villagómez-Aranda et al. reported a comparative DNA methylation study in tobacco.

  1. What are the genetic backgrounds of the two tobacco lines compared? How different are they? Can the different phenotypes observed mainly caused by genetic differences instead of DNA methylation?
  2. More QC info of the WGBS libraries needed to assess the quality of the DNA methylation results.
    1. What is the conversion rate? Did the authors use lambda control during bisulfite conversion? Since the observed DNA methylation differences are so small on the whole genome scale, it is very important to know the bisulfite conversion rate.
    2. What is the genomic coverage of these libraries (at least one read)?
    3. What are percentages of C covered in the genome for each library?
    4. How did the author process the C/T SNPs between these two tobacco lines?
  3. The author should also try to define DMR between these two lines.

Author Response

All the address to reviewer 1 comments are included in the attached file. Thank you very much for your helpful comments.

Reviewer 2 Report

The manuscript submitted by Villagómez-Aranda etc. described the 5mC methylome of a hydrogen peroxidase overproducing tobacco plant. Here, they have linked DNA methylation and H2O2 in a transgenic line tobacco plant (L8). Overall, the manuscript presented a well-designed and well-executed study.

In Introduction

L42-43: there are several other DNA methylation types, such as 7mG, are emerging, although not as prevalent and well-studied as 4/5mC and 6mA.

In Result

Fig2 A, since the authors have sequenced 3 biological replicates, it should contain an error bar for the bar plot.

Result 2.5, Gene body methylation alteration normally will lead to alternative splicing, not differential expression. It’s may worth trying some mRNA alternative splicing analysis for these genes. Or at least this result should be further discussed.

In Discussion

Is the tobacco transgenic system needs tissue culture? If so, transgenic process itself may cause some DNA methylation differences, as well established in rice (Hume Stroud, elife, 2013), maize (Zhaoxue Han, Genetics, 2018), oil palm (Meilina Ong-Abdullah, Nature, 2015). If tissue culture has been used in transgenic, then tissue culture induced DNA methylation changes also has to be considered and discussed.

In Method

WGBS data needs a negative control to evaluate bisulfite conversion rate. In plants, chloroplast genome is virtually none methylated. Thus the authors can add methylation level of chloroplast genome as an indicator of conversion rate.

Further point:

Since hydrogen peroxidase itself can’t cause DNA methylation alteration, the authors should check expression level of DNA methyltransferases (such as CMT3) in the transgenic plant.

Author Response

Address to all reviewer 2 comments are included in the attached file. Thank you very much for your helpful comments.

Round 2

Reviewer 1 Report

The authors have addressed all my concerns.